# Thermal Properties and the Prospects of Thermal Energy Storage of Mg–25%Cu–15%Zn Eutectic Alloy as Phase Change Material

**DOI:** 10.3390/ma14123296

**Published:** 2021-06-15

**Authors:** Zheng Sun, Linfeng Li, Xiaomin Cheng, Jiaoqun Zhu, Yuanyuan Li, Weibing Zhou

**Affiliations:** School of Materials Science and Engineering, Wuhan University of Technology, Wuhan 430070, China; sunzheng400@163.com (Z.S.); lilinfeng0514@whut.edu.cn (L.L.); zhujiaoq@whut.edu.cn (J.Z.); yyli@whut.edu.cn (Y.L.); jsyczwb@whut.edu.cn (W.Z.)

**Keywords:** magnesium-based alloy, latent heat storage, high temperature PCM, thermal reliability

## Abstract

This study focuses on the characterization of eutectic alloy, Mg–25%Cu–15%Zn with a phase change temperature of 452.6 °C, as a phase change material (PCM) for thermal energy storage (TES). The phase composition, microstructure, phase change temperature and enthalpy of the alloy were investigated after 100, 200, 400 and 500 thermal cycles. The results indicate that no considerable phase transformation and structural change occurred, and only a small decrease in phase transition temperature and enthalpy appeared in the alloy after 500 thermal cycles, which implied that the Mg–25%Cu–15%Zn eutectic alloy had thermal reliability with respect to repeated thermal cycling, which can provide a theoretical basis for industrial application. Thermal expansion and thermal conductivity of the alloy between room temperature and melting temperature were also determined. The thermophysical properties demonstrated that the Mg–25%Cu–15%Zn eutectic alloy can be considered a potential PCM for TES.

## 1. Introduction

Thermal energy storage (TES) can deal with the mismatch between intermittent energy supply and demand by storing heat and cold for later use. Therefore, TES is drawing great research interest for various engineering applications, such as concentrating solar power (CSP) [1], waste heat recovery [2], building energy conservation [3] and automotive engine cooling [4]. Compared with other TES methods, the phase change materials (PCMs) for latent heat storage have attracted widespread attention due to high heat storage density per unit volume and good reversibility during the heat absorption and liberation process [5,6].

When choosing a phase change energy storage material, its thermal properties, such as working temperature, heat capacity, thermal conductivity and thermal reliability, are often valued. In addition, there have been some experimental and numerical studies based on the application of the TES system [7,8]. Hoshi et al. [9] divided the PCMs related to the CSP system into a low temperature range below 220 °C, a medium range up to 420 °C, and a high range over 420 °C. It is believed that PCMs with a higher phase change temperature generally allow for higher operating temperature and energy storage, which can considerably improve the system-level efficiency of current CSP plants. Candidates for high temperature phase change materials generally include inorganic salts and metals, either in pure form or as eutectic mixtures [10,11]. Among them, metallic alloys possess high thermal conductivity and low overcooling degree, which means they could be promising high temperature PCMs.

Recently, certain binary and ternary eutectics containing Al, Cu, Mg, Si and Zn, especially Al-based, alloys have been investigated for heat storage applications [12,13]. For instance, Wang et al. developed a new heat storage alloy, Al–12%Si (wt.), which can be used as a heat storage medium for high temperature space heaters [14]. Al–12%Si alloy, identified as an effective PCM, is also used in different shell-tube latent heat exchangers [15,16]. Al–40%Si–15%Fe (at.%) alloy, synthesized by the hot pressure method, has a heat of fusion of 865 kJ/kg with a melting temperature of about 876 °C [17]. Sun et al. [18] found that Al–34%Mg–6%Zn (wt.%) alloy appeared to have good thermal stability over 1000 melting and solidification cycles, and stainless steel 304L can be a relatively suitable encapsulation material for the Al–34%Mg–6%Zn alloy. However, aluminum melt is highly corrosive to candidate packaging materials that are based on iron [19], which is the main reason that the long-term and large-scale application of Al-based alloys is limited.

Magnesium has similar thermophysical properties to aluminum, such as melting point, heat of fusion, specific heat, volume expansion coefficient and so on. Besides, the Mg-Fe system is thermodynamically stable and immiscible in the concerned temperature range of 400–600 °C, according to the phase diagram [20]. With these considerations, Mg-based alloys may be considered promising PCMs for TES applications. Based on previous studies involving the mechanical and thermal properties of Mg-based alloys [21,22,23], there has been research on magnesium based alloys as PCMs for thermal energy storage in recent years [24,25]. The Mg–Cu–Zn alloy system has been reported before, but the above mentioned systems were focused on mechanical properties [26] and there was no report on its application as a PCM. In this work, we focused on the thermal stability of Mg–25%Cu–15%Zn (wt.%) eutectic alloy as a PCM for TES applications in the temperature range of interest, from 400 °C to 500 °C. The phase composition, microstructure, phase change temperature and enthalpy of the alloy after 500 heating and cooling cycles were studied. Moreover, the significant thermal properties of Mg–25%Cu–15%Zn alloy were also determined.

## 2. Materials and Methods

### 2.1. Materials and Preparation

Mg–25%Cu–15%Zn eutectic alloy, with the main composition of Mg 59.88, Cu 25.50 and Zn 14.62 (wt%), was prepared with magnesium ingot with a purity of 99.98%, copper ingot with a purity of 99.99% and zinc ingot with a purity of 99.95%. The alloy was melted (600 g melt) in a graphite crucible using a vertical resistance furnace under the protection of a RJ-2 flux (Hebi Taihang Technology., Ltd. Hebi, China). An atmosphere of high purity argon (99.999%) was provided to strictly prevent the oxidation of samples during preparation. A sample with a diameter of 20 mm was cast in an iron mold.

(1) Measurement and weighing. The metal raw materials were cut into blocks of the appropriate size with a numerical control band sawing machine. The oxide film on the surface was removed by angle grinder and was then weighed. The distribution ratio of each group was calculated according to the total amount of 1 kg of alloy. The ratios of Mg, Cu and Zn were added to burning loss, which were 2.0%, 0.5% and 3.0% (wt.%), respectively.

(2) Pretreatment. The inner wall of the graphite crucible, the front end of the stirring rod and the inner wall of the cast iron mold will make contact with the Mg alloy melt, so it was necessary to evenly brush the release agent with a thickness of about 0.5 mm, and put it into the resistance furnace to bake and dry at 200 °C. The melting tongs, skimming scoops and weighed metal raw materials were also baked and dried at 200 °C. The graphite crucible with a cover was put into the smelting furnace and preheated to a dark red state at 450–500 °C.

(3) Smelting preparation. A small amount of RJ-2 covering agent was evenly spread on the bottom of the crucible. First, the preheated Mg block was put into the crucible, and the covering agent accounting for 2–5% of the total weight of the alloy was added to cover the Mg block. Then, the graphite crucible was heated to 700 °C and the Mg was completely melted. Then, the Cu block was added to keep the block in the immersion state, the temperature was raised to 720–750 °C for 30 min, and the Zn block was added after the Cu block was completely melted. The furnace temperature was adjusted to 560–600 °C, a small amount of covering agent was sprinkled on the surface of the alloy liquid, and the alloy melt was stirred slowly with a stirring rod at a uniform speed; finally, refining agent was added and fully stirred to refine the alloy liquid.

(4) Casting. The cover of the crucible was opened, the graphite crucible was taken out with melting tongs, the preheated cast iron mold was taken out, the slag on the surface of the alloy liquid was removed with a skimming spoon, the alloy liquid was poured into the mold slowly and evenly, and was demoulded after natural cooling in the air to obtain the as-cast Mg–Cu–Zn alloy with a diameter of 30 mm and a height of 130 mm.

### 2.2. Thermal Cycling Test

Pipes of diameter 42 mm, thickness 4 mm and length 50 mm, made of 304L stainless steel, were filled with the cylindrical alloy specimen of diameter 20 mm and length 45 mm. Both ends of the pipe were sealed with argon arc welding. The pipes were placed in a vertical resistance furnace in the open atmosphere, heated to 600 °C for 30 min, and then furnace cooled to 400 °C for another 30 min. Both the heating and cooling scanning rates were 10 K/min. The same heating and cooling process was repeated 100, 200, 400, 500 times, respectively.

### 2.3. Analysis Methods

Alloy samples in powder form were analyzed by the X-ray diffraction technique using an X-ray powder diffractometer (XRD, D8 ADVANCE, Bruker, Karlsruhe, Germany) with Cu-Kα radiation at 40 KV and 30 mA. The alloy powders were packed in a rectangular aluminum reduction cell (20 × 20 mm, thickness 0.15 cm) and measured in the 5–80° diffraction angle region. Electron probe micro-analysis (EPMA, JXA-8230, JEOL, Akishima, Japan) was conducted to characterize the as-casted microstructures. A quantitative energy dispersive spectrometer (EDS, INCA X-ACT, Oxford Instruments, Abingdon, UK) system attached to the EPMA instrument was used to analyze the chemical composition of the phases. In this study, EPMA was conducted in the spot mode with an accelerating voltage of 10 kV and a probe current of 10 nA. The beam diameter was in the range of 1 to 2 µm.

Differential scanning calorimetry (DSC, STA449C/3/G, Netzsch, Germany) analyses were carried out at 25–500 °C with a constant heating and cooling rate of 10 K/min in an argon atmosphere. A pushrod type dilatometer (DIL 402C, Netzsch, Germany) was used to measure the linear thermal expansion coefficient and density values of alloy in the temperature range of 25 to 450 °C at a heating rate of 5 K/min. The sample for the dilatometry test was machined to dimensions of Φ5 × 23 mm.

The thermal diffusivity of 8 mm × 8 mm × 2 mm bulk samples was measured by laser flash-method (LFA 457, Netzsch, Germany) in the temperature range of 25 to 400 °C. The test temperature was set to 50 °C and at least three measurements were taken at each test point. The specific heat capacity was calculated using the Neumann-Kopp rule and published data [27]. The thermal conductivity can be obtained using the following equation:*λ* = *a*·*ρ*·*c_p,_*(1)
where *a* is the thermal diffusivity, which has been measured by laser flash-method, *ρ* is the density and *c_p_* is the specific heat capacity at constant pressure.

## 3. Results

### 3.1. Structural Analysis

Figure 1 shows X-ray diffraction patterns of Mg–25%Cu–15%Zn alloy after 100, 200, 400 and 500 thermal cycles. The sharp diffraction peaks come from the mixture of hexagonal α-Mg, body-centered cubic Mg_7_(Zn,Cu)_3_ and tetragonal MgCuZn phases, indicating that there is no obvious change in thermal properties after 500 cycles. With an increased number of thermal cycles, the diffraction intensities of the α-Mg and Mg_7_(Zn,Cu)_3_ phases are continuously strengthened. This indicates that the degree of crystallization of α-Mg and Mg_7_(Zn,Cu)_3_ phases maybe improve with increasing number of thermal cycles.

Figure 2 shows electron probe micro-analysis (EPMA) images of as-cast Mg–25%Cu–15%Zn alloys before and after 500 thermal cycles. Table 1 shows the composition of the intermetallic phases exhibited in Figure 2, obtained from energy dispersive spectrometer (EDS) analysis. There is no significant change in the as-cast alloy before and after thermal cycling. The microstructure of the alloy is a typical cellular eutectic structure, containing dark dendrites homogeneously embedded in the gray eutectic matrix.

Based on the results from the XRD and EDS analysis, it is believed that the microscale primary dendrite is α-Mg solid solution and the cellular eutectic matrix consists of a mixture of α-Mg, Mg_7_(Zn,Cu)_3_ and MgCuZn phases. The similar microstructure and phase composition of Mg_90_Cu_5_Zn_5_ (at.%) alloy has already been reported in Reference [28]. The micrograph in Figure 2a presents the micron-scale α-Mg dendrites with a size of 3–20 μm. After 500 thermal cycles, the size of α-Mg in Figure 2b increases to 6–25 μm and the grain boundary area tend to slightly decrease. The growth of dendrites causes the diffraction intensities of α-Mg phase to enhance, which is coordinated with the results of XRD analysis. In addition, it is observed that the amount of dendrites is substantially reduced, which is attributed to the dendrites (such as α-Mg solid solution) being desolved after thermal cycles, and the smaller grains disappearing, while the larger grains continue to grow.

### 3.2. Phase Change Temperatures and Enthalpies

Figure 3 and Table 2 reveal the measured phase change temperatures and enthalpies of Mg–25%Cu–15%Zn alloy after 100, 200, 400 and 500 thermal cycles. As shown in the DSC curves in Figure 3, there is only one endothermic and one exothermic peak on the DSC curve, which is consistent with the phase diagram showing that there is only one melting point in the temperature range of 400–500 °C [20]; the phase diagram can be seen in Figure 4. Therefore, it indicates that the eutectic composition of Mg–25%Cu–15%Zn alloy is stable after 500 thermal cycles. The data in Table 2 show that the melting and freezing temperatures are determined to be 452.6 °C and 448.2 °C for the alloy before thermal cycling. As for the onset melting temperature, the result agrees with the value in the literature [29]. The melting enthalpy value of 177.5 J/g is below the reported data given by the authors (254.0 J/g), It may be caused by different experimental preparation conditions, heating and cooling rates and calorimeter accuracies. Herein, the melting enthalpy of Mg/Zn alloy with 155 J/g was given in Reference [24], so the melting enthalpy of eutectic alloy in this study is more reliable by comparison.

After 100, 200, 400 and 500 thermal cycles, the melting temperature value changes by −0.4, 1.1, 1.7 and −1.3 °C, and the freezing temperature changes by −0.1, −1.4, −1.3 and −2.0 °C, respectively. On the other hand, the enthalpies of melting and freezing are found to be 177.5 J/g and 171.4 J/g for alloy before thermal cycling. The enthalpy value of melting of the alloy changes by −0.5%, −3.9%, −5.8% and −6.7% and the enthalpy value of freezing changes by −3.6%, −4.6%, −7.8% and −7.5% after repeated 100, 200, 400 and 500 thermal cycles. The decrease of latent heat after cycling might be attributed to the loss of Zn, because Zn has a lower melting point [30]. The transformation temperature of the alloy has no obvious change. The latent heat of the Mg–25%Cu–15%Zn alloy gradually decreases with the increasing number of thermal cycles. Similar results with Al–34%Mg–6%Zn alloy have been reported by Sun et al. [18]. In addition, after 500 thermal cycles, the Mg–25%Cu–15%Zn alloy possesses a small overcooling that varies between 4.1 and 7.4 °C.

### 3.3. Thermal Expansion

The coefficient of thermal expansion is an important parameter related to whether Mg–25%Cu–15%Zn eutectic alloy can be used as a phase change material, so the application of thermal expansion analysis and the thermal expansion test is very important. Figure 5 shows the results of the linear expansion coefficient, relative elongation and density of this alloy. As shown in the curves from Figure 5a, the linear expansion coefficient and relative elongation exhibit an obvious increase from room temperature up to melting temperature. Then, the phase transition occurs at 442.0 °C, which is lower than 452.6 °C in the DSC measurements. This difference in onset temperature may be attributed to the different heating rate and calorimeter accuracies, and can also be related to the different sample masses used for each analysis. Besides, a change in the slope of the temperature dependence of the linear expansion coefficient is perceptible at high temperatures and almost linear processes of the relative elongation are identifiable. In general, the linear expansion coefficient increases rapidly with the increase of temperature, and tends to be constant above the Debye characteristic temperature. This experiment conforms to this rule.

The density at high temperature is calculated by using the relation [31]:*ρ* = *ρ*_0_ (1 + *ΔL/L*_0_)^−3,^(2)
where *ρ_0_* is the density of the alloy at 25 °C, which has been measured by the Archimedes drainage method, and *ΔL/L_0_* is the relative elongation, which has been measured with a pushrod type dilatometer.

Based on Equation (2) and the data above, a linear fitting curve of density is also obtained in Figure 5b. The experimental value of the density at room temperature of Mg–25%Cu–15%Zn alloy is measured to be 2571 kg/m^3^ by the Archimedes method. This value is lower than that found in the Mg–25%Cu–15%Zn (2800 kg/m^3^) [29], which may be attributed to a few pores and inclusions introduced during the process of melting and casting. Thus, it can be estimated that the rate of volume change of the alloy is less than 3.56% from room temperature to 442 °C, the density decreases by 3.44%, and the latent heat capacity is around 0.46 MJ/m^3^. Besides, the linear thermal expansion coefficient, relative elongation and density can be described as a function of the temperature by the following equations:*α* (10^−6^/K) = 15.878 + 0.138 T − 7.195 × 10^−4^ T^2^ + 1.763 × 10^−6^ T^3^ − 1.595 × 10^−9^ T^4^, 25 °C ≤ T ≤ 442 °C(3)
*ΔL/L*_o_*=* −8.795 × 10^−4^ + 2.875 × 10^−5^ T, 25 °C ≤ T ≤ 442 °C(4)
*ρ* (kg/m^3^) = 2576 − 2138 × 10^−4^ T, 25 °C ≤ T ≤ 442 °C.(5)

### 3.4. Thermal Conductivity

The thermal conductivity of PCM is a crucial factor for the heat transfer coefficient of TES systems. Equation (1) reveals the relationship between thermal conductivity and thermal diffusivity, density and specific heat capacity. According to the obtained parameters, the thermal conductivity can be calculated. The results of the measured thermal diffusivity and calculated specific heat capacity of Mg–25%Cu–15%Zn alloy are presented in Figure 6. The thermal diffusivity shows a minor fluctuation from room temperature to 250 °C. Then, it decreases slightly with increasing temperature; this may be due to the strong vibration of atoms, which provides a great opportunity for scattering electrons [32]. The specific heat curve of linear growth is shown below the phase transition temperature in Figure 6b. The specific heat value passes from 0.77 kJ·kg^−1^·K^−1^ at 25 °C up to 0.94 kJ·kg^−1^·K^−1^ at 400 °C. This value is higher than the 0.84 kJ·kg^−1^K^−1^ of Mg–51%Zn alloy reported in the literature [24] and lower than the specific heat value of 1.20 kJ·kg^−1^·K^−1^ of pure Mg [32].

The temperature dependence of the thermal conductivity for the Mg–25%Cu–15%Zn alloy is shown in Figure 7. The thermal conductivity generally increases linearly with increasing temperature. There is a slight variation in the slope of the temperature dependence of the thermal conductivity at 350 °C, which is in agreement with that of thermal diffusivity. The experimental data showed that the thermal conductivity increases from a value of 120 W·m^−1^·K^−1^ at 25 °C, up to 141 W·m^−1^·K^−1^ at 400 °C.

## 4. Conclusions

In this paper, thermal reliability and significant thermal properties of Mg–25%Cu–15%Zn eutectic alloy as a latent heat energy storage material for CSP applications are reported. Firstly, there is no considerable phase transformation nor structural change in the alloy after 500 thermal cycles. The change in melting and freezing temperatures for the alloy were −1.3 °C and −2.0 °C, and the melting and freezing enthalpies decreased by 6.65% and 7.53% after 500 thermal cycles. The thermal cycling test indicated that the Mg–25%Cu–15%Zn alloy has good thermal stability over long-term use. Furthermore, from the results of the thermal expansion and thermal conductivity tests, the latent heat capacity is 0.46 MJ/L, the volume growth rate of the alloy is less than 3.56% before melting, and the value of thermal conductivity is from 120 to 141 W/mK in the range 25–400 °C. The results show that Mg–25%Cu–15%Zn eutectic alloy has potential application in CSP energy storage materials. In addition, the compatibility of Mg based alloys with packaging materials is also an important issue, which will be further studied in a future paper.

## Figures and Tables

**Figure 1 materials-14-03296-f001:**
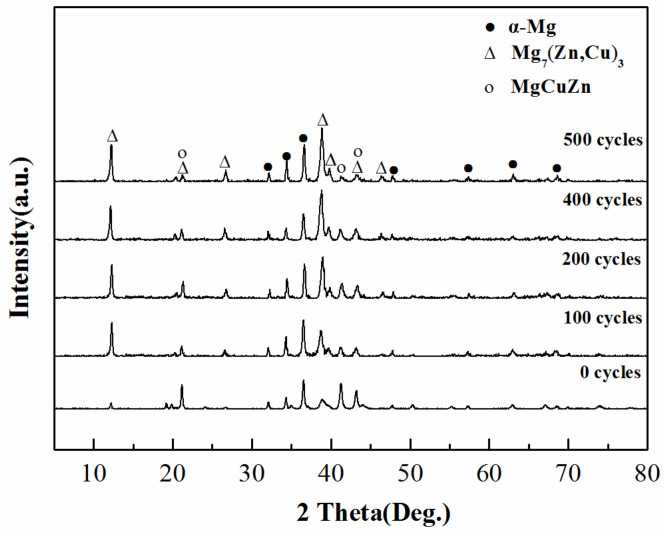
XRD spectrum of as-cast Mg-25%Cu-15%Zn alloy after different thermal cycles.

**Figure 2 materials-14-03296-f002:**
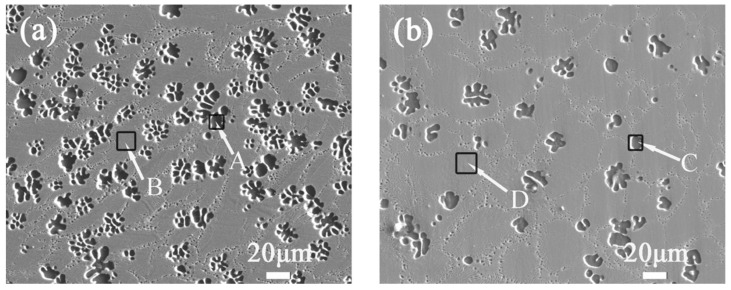
EPMA images of as-cast Mg–25%Cu–15%Zn alloys (**a**) before and (**b**) after 500 thermal cycles.

**Figure 3 materials-14-03296-f003:**
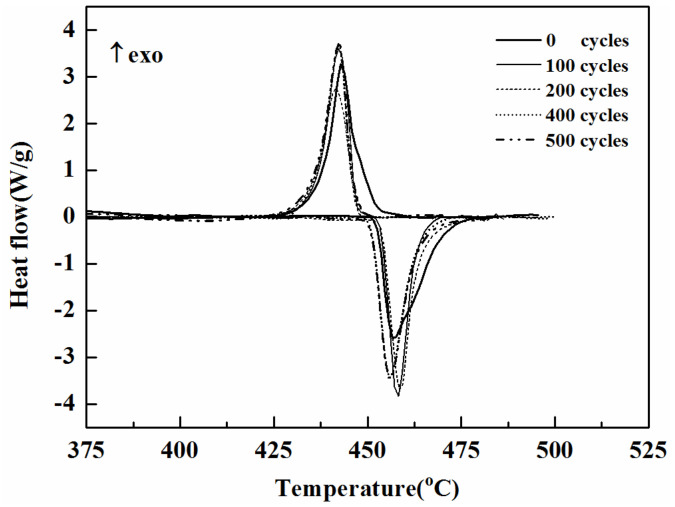
DSC curves of Mg–25%Cu–15%Zn alloy after different thermal cycles.

**Figure 4 materials-14-03296-f004:**
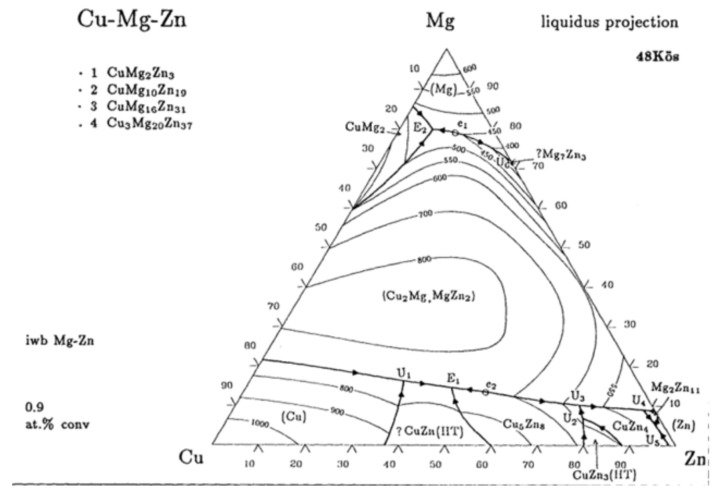
Phase diagram of Cu–Mg–Zn alloy [20].

**Figure 5 materials-14-03296-f005:**
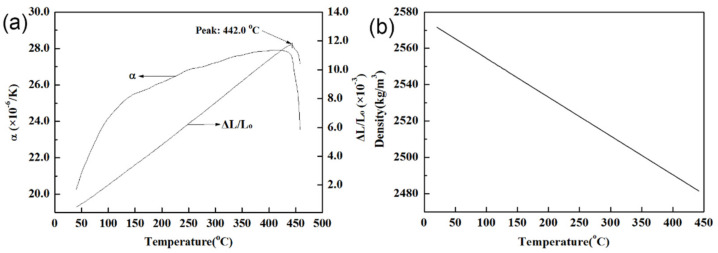
(**a**) Linear expansion coefficient and relative elongation *vs* temperature, (**b**) linear fitting curve of density *vs* temperature.

**Figure 6 materials-14-03296-f006:**
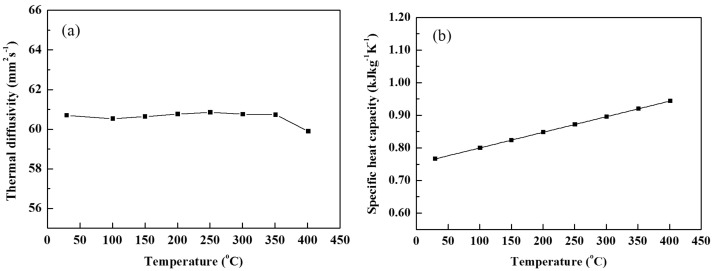
Thermal diffusivity (**a**) and specific heat capacity (**b**) of Mg–25%Cu–15%Zn alloy *vs* temperature.

**Figure 7 materials-14-03296-f007:**
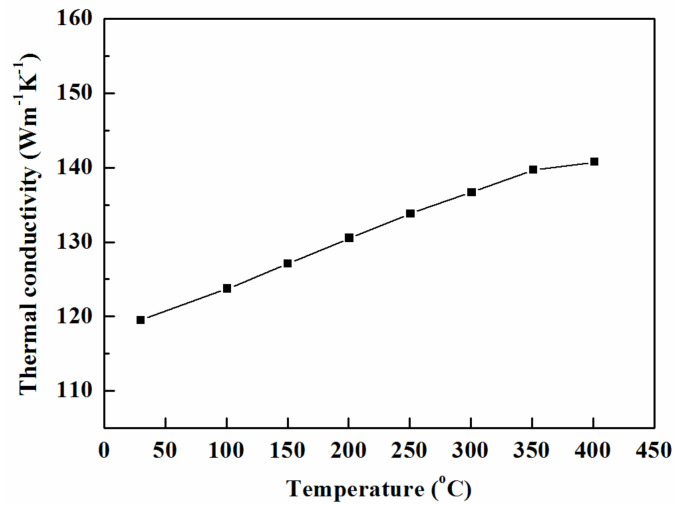
Thermal conductivity of Mg–25%Cu–15%Zn alloy vs temperature.

**Table 1 materials-14-03296-t001:** Composition of the intermetallic phases of 4 regions in Figure 2.

Phase	A	B	C	D
Mg	98.56	71.11	98.21	71.35
Cu	0.55	17.69	0.48	19.17
Zn	0.89	11.20	1.31	9.48
Closest phase	α-Mg	α-Mg + Mg_7_(Zn,Cu)_3_ + MgCuZn	α-Mg	α-Mg + Mg_7_(Zn,Cu)_3_ + MgCuZn

**Table 2 materials-14-03296-t002:** Various thermodynamic properties of Mg–25%Cu–15%Zn alloy after different thermal cycles.

Number of Cycles	Melting Temperature /(°C)		Freezing Temperature /(°C)		Overcooling Degree /(°C)	Melting Enthalpy /(J/g)	Freezing Enthalpy /(J/g)
	Onset	End	Onset	End	ΔT	ΔH_m_	ΔH_f_
0	452.6	471.8	448.2	437.7	4.4	177.5	171.4
500	451.3	462.9	446.2	433.6	5.1	165.7	158.5

## Data Availability

Not applicable.

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
