# Peer review of "Thermal Properties and the Prospects of Thermal Energy Storage of Mg–25%Cu–15%Zn Eutectic Alloy as Phase Change Material"

_materials, 2021, doi:10.3390/ma14123296_

Round 1

Reviewer 1 Report

The subject is not innovative and the English language must be improved. Some paragraphs are very difficult to understand.

1/ The state of the art must be updated : numerous studies have been done on this subject since 2014.

2/ The authors mention a phase diagram all along the paper from which the ternary eutectic material has been selected but neither figure of this phase diagram with the material of interest is shown to facilitate the reading nor its right chemical formulae. Moreover, it is impossible to obtain some of the information from a phase diagram as claimed by the authors such as, for instance, l. 146-148 : the stability of a material over the thermal cycle number...).

3/ The Materials and Methods section must be improved : too much information is missing to be able to check the reliability and the relevancy of the presented experimental results afterwards.

4/ The interpretation of the results should be completely revised...

Author Response

1/ response: Thanks for your careful review. After reading your comment, I realized that Mg-based alloys as PCM for thermal energy storage have attracted more attention since 2014, so I changed the expression as shown in line 59. “Whereas, most currently studies concerned about the mechanical and thermal properties of Mg-based alloys as structure materials [19-21], and there have been only a few researches on the Mg-based alloys as PCM for thermal energy storage [22, 23].” is replaced by “Based on the previous studies involving the mechanical and thermal properties of Mg-based alloys [19-21], there have been researches on magnesium based alloys as PCM for thermal energy storage [22, 23].”

2/ response: Thanks for your careful review. The phase diagram as shown in below. It can be seen from the phase diagram that there is only one melting point in the temperature range of 400-500 °C, while there is only one endothermic peak on the DSC curve in this paper, so it is consistent with the conclusion of reference. In order to let readers understand the research more clearly, I have added a description here. Please see the revised version of the manuscript.

3/ response: I have added more information about the detailed method of alloy preparation in line 76. And you can find in attachment.

4/ response: I realized that the interpretation of result should be modified, so I revised it as you can see in new manuscript.

Reviewer 2 Report

The manuscript presents the characterization of Mg–25%Cu–15%Zn eutectic alloy as high temperature phase change material (PCM) for thermal energy storage (TES) and the results are detailed. The discussion and conclusion reveal an interesting application for PCMs.  The study covers the thermo-physical properties study of the Mg alloy material as PCMs.  Some recommendations are still suggested in order to improve the manuscript:

Section 1 its well-structured and covers a generic literature review on Al and Mg studies for PCM applications. This section could be consolidated at an initial stage with information regarding active and passive TES systems for a range of applications. For instance, for active systems, additional references should be added, example of experimental and numerical approach can be seen below:

S Farah, M Liu, W Saman. Numerical investigation of phase change material thermal storage for space cooling. Applied Energy. 239 (2019) 526-535.

M Iten, S Liu. Experimental Study on the Performance of RT 25 to be Used as Ambient Energy Storage. Energy Procedia. 70 (2015) 229-240.

In Section 2.3, please justify why a rate of 10 K/min has been used in the DSC testing also please provide a brief justification for the selected temperature range of the analysis: 25-500 ◦C. In section 1, it is mentioned that the studied PCM is intended for application ranging from 400 ◦C to 600◦C.  Moreover, what was the reason for a different rate of 5K/min for the other testing. This explanation is relevant as all results: phase change temperatures and enthalpies; thermal expansion and thermal conductivity are presented only until 400◦C.

Author Response

1/ response: Thanks for improving my Introduction part, I have added the references that you recommended as shown in para. 2 of Introduction.

2/ response: Thanks for your question. In fact, the instruments (DSC) used to measure the phase change temperature and melting point are different from those used to measure the linear thermal expansion coefficient and density values of alloy, so the heating rate is different. In addition, as for “PCM is intended for application ranging from 400 to 600 °C”, at the first, according to the phase diagram, there are no endothermic and exothermic peaks between 500 - 600 °C, so I removed the DSC curve of this range. I respect your comments, to make the conclusion more convincing, I decided to change my expression as you can see in line 63.

Reviewer 3 Report

Please carefully read my comments and make the changes and corrections suggested in the attached document

Author Response

I deeply agree with your view that 500 cycles do not ensure the reliability of the material, so I changed the expression, as you can see in Abstract. In addition, I revised the language, please check it in new manuscript. Detailed response please find in attachment.

1/ response: Thank you for your reminding, I have changed the way of description according to your suggestion.

2/ response: The problems in Line 53, Line 58-60, Line 63-64 have been corrected.

Line 70, 75: I have rewritten the experimental part as you can see in revised version.

Line 82: Yes, I’ve corrected it.

Line 86: Thanks for reminding. I have changed "as-cast" in line 86 to "as-casted"

Line 85-89: Thank you for your suggestion, I've described the XRD measurement in a different way.

Line 106: Relative elongation has been measured by pushrod type dilatometer, the determination methods of the related parameters have been supplemented.

Line 113: Thank you for your reminding, I’ve corrected it.

Line 133: Thanks for reminding, I’ve corrected it.

Line 137-140: The reason for the decrease of dendrite number after cycling is that dendrites (such as α-Mg solid solution) are desolved and the smaller grains disappear, while the larger grains continue to grow. I will add my description to the manuscript.

Line 152: The alloy in the related literature is the same alloy prepared in this paper. This part is described in more detail.

Line 161-167: Thanks for your suggestion, I've deleted the redundant data.

Line 172: Thank you for your reminding, I’ve corrected it.

Line 182: The coefficient of linear expansion is obtained from the test data and the fitting curve fitting formula.

Line 187: Thanks for reminding, I’ve corrected it.

Line 188-189: In general, the coefficient of linear expansion increases rapidly with the increase of temperature and tends to be constant above the Debye characteristic temperature. This experiment conforms to this rule.

Line 195: The data can be seen in Table 4 summary of heat of fusion measurements, page 326, you can find the figure in attachment.

Line 202: Equation (2) is the definition of thermal diffusivity, and the thermal conductivity can be calculated according to the experimental measurement of thermal diffusivity, density and specific heat capacity; Equation (3) is the coefficient of linear expansion, which is obtained from the experimental data of numerical fitting.

Line 205: The specific heat capacity at constant pressure, which can be calculated using the Neumann-Kopp rule and published data [27].

Cp=C1X1+C2X2+...+CnXn

Where C1, C2 And Cn are the specific heat capacities of each component, J/(g·K); X1, X2 and Xn is the mass percentage of each component in the alloy.

The thermal diffusivity has been measured by laser flash-method, the density calculated by Equation (1). Based on the values of thermal diffusivity, density and specific heat capacity obtained, the thermal conductivity can be calculated using the Equation (2).

Line 221: The mistake in line 221 has been corrected.

Line 223-224: The thermal conductivity in line 223 to 224 is the experimental data of Mg–25%Cu–15%Zn alloy, this experiment is a part of this paper. I've described it more accurately in the paper.

Line 238: The mistake in line 238 has been corrected.

Line 241: The mistake in line 241 has been corrected.

Reviewer 4 Report

The manuscript focusses on the thermal properties and reliability of Mg–25%Cu–15%Zn Eutectic Alloy as PCM. The results of this work demonstrate the potential use of this compound as PCM, nevertheless, there are some questions that should be addressed:

  • In the abstract section, why do you use the term “high temperature-resistant eutectic alloy”? Does this material have an unusual temperature-resistance? At least in this paper you demonstrate the reliability in its typical working temperature range, but not the maximum temperature resistance.
  • It would be interesting to see the phase change temperature in the abstract because it is the main characteristic of a PCM.
  • On line 41, “In addition, there were some experimental and numerical studies based on the application of TEs system [7, 8].” This sentence is out of context within the text.
  • On line 48, “Inorganic salts and metals either in pure form or as eutectic mixtures [10, 11].” Incomplete sentence.
  • In order to highlight the relevance of the manuscript, the novelty of the work must be clearly indicated in the introduction section: Has this system ever been reported before? or if it has been reported, Has this system ever been focused on its application as a PCM?
  • On line 162, “Set the test temperature to 50 °C and make at least three measurements at each test 162 point.”: Is this sentence referred to the laser flash-method or to the dilatometer? Clarify this point.
  • The calculation of the density at high temperature (Eq.1) should be located just after the explanation of the dilatometry test.
  • On line 179, the conclusion: “….indicating that there is no obvious phase transition after 500 cycles.” It is a bit confusing taking into account that you are talking about cycling through the phase transition of the alloy.
  • On line 219 you have two repeated explanations regarding the differences between your latent heat and the one reported in literature.
  • Figure 3: Maybe it is due to the pdf generation process, but if possible try to improve the quality of this figure.
  • On line 237: What can be the reason of the decrease of latent heat after cycling? TGA (thermogravimetric analysis) of the alloy up to 700ºC is required and could complete your work. Loss of Zn could be a reason.
  • On line 247: “The coefficient of thermal expansion is an important parameter related to whether Mg–25%Cu–15%Zn 248 eutectic alloy can be used as phase change material, so the application of thermal expansion analysis and thermal expansion test is very important.” Why is this coefficient  very important to determine if the material can be used as PCM?
  • On line 253: “Then, the phase transition occurs at 442.0 °C, which is lower than 452.6 °C in the DSC measurements. This difference in onset temperature may be attributed to the different heating rate and calorimeter accuracies.” The different onset Temperature can be also related with the different sample masses used for each analysis (effect of sample mass in thermal properties is reported in literature).
  • The volume change from room temperature to 442ºC has been analyzed. What about the volume change due to the phase change (very important parameter overall taking into account that your PCM is sealed in a container), and density of liquid phase? It will be very helpful to have the volume expansion of the phase change. (recommendation)
  • On line 289: “This value is higher than the result of Mg–51%Zn alloy reported 289 in the literature [24] and lower than the specific heat value of pure Mg [30].” Specify the values in the text.
  • If the material is going to be applied as PCM the working temperature range will cover not only the solid phase, but also the liquid phase. Could you test the thermal conductivity of liquid phase? And do you have any information or have you tested the volume expansion due to the phase change? (recommendation)

Congratulation for your work, and I hope the authors find these recommendations useful for improving the understanding of their work.

Author Response

Thanks for your careful review, I do think my manuscript can be improved by your comments, and I have make some revision according to your comments.

  • In the abstract section, why do you use the term “high temperature-resistant eutectic alloy”? Does this material have an unusual temperature-resistance? At least in this paper you demonstrate the reliability in its typical working temperature range, but not the maximum temperature resistance.

Answer: Thanks for reminding, I have deleted the term “high temperature-resistant eutectic alloy” to make my manuscript more convincing as you can see in line 10.

  • It would be interesting to see the phase change temperature in the abstract because it is the main characteristic of a PCM.

Answer: Thank you for your suggestion, I have added the information of phase change temperature in abstract in line 14.

  • On line 41, “In addition, there were some experimental and numerical studies based on the application of TEs system [7, 8].” This sentence is out of context within the text.

Answer: In fact, in the first peer expert review, reviewer #2 proposed that reference [7, 8] will be helpful to improve the expression of the Introduction. The detailed comments as follow: “Section 1 its well-structured and covers a generic literature review on Al and Mg studies for PCM applications. This section could be consolidated at an initial stage with information regarding active and passive TES systems for a range of applications. For instance, for active systems, additional references should be added, example of experimental and numerical approach can be seen below: 1. S Farah, M Liu, W Saman. Numerical investigation of phase change material thermal storage for space cooling. Applied Energy. 239 (2019) 526-535. 2. M Iten, S Liu. Experimental Study on the Performance of RT 25 to be Used as Ambient Energy Storage. Energy Procedia. 70 (2015) 229-240.”

  • On line 48, “Inorganic salts and metals either in pure form or as eutectic mixtures [10, 11].” Incomplete sentence.

Answer: The sentence should be modified as “Candidates for high temperature phase change materials generally include inorganic salts and metals either in pure form or as eutectic mixtures”, and I have corrected it as in line 40.

  • In order to highlight the relevance of the manuscript, the novelty of the work must be clearly indicated in the introduction section: Has this system ever been reported before? or if it has been reported, has this system ever been focused on its application as a PCM?

Answer: Thanks for your suggestion, Mg–Cu–Zn alloy system has been reported before, but the above mentioned systems were focused on mechanical properties, and there was no report on its application as PCM. I have added this description in line 65.

  • On line 162, “Set the test temperature to 50 °C and make at least three measurements at each test 162 point.”: Is this sentence referred to the laser flash-method or to the dilatometer? Clarify this point.

Answer: This sentence referred to laser flash-method, and I had made an expression in line 133 “The thermal diffusivity of 8 mm × 8 mm × 2 mm bulk samples was measured by laser flash-method (LFA 457) in the temperature range of 25-400 °C.”

  • The calculation of the density at high temperature (Eq.1) should be located just after the explanation of the dilatometry test.

Answer: Thanks for your advice, I have changed the location of Eq.1 according to your advice.

  • On line 179, the conclusion: “….indicating that there is no obvious phase transition after 500 cycles.” It is a bit confusing taking into account that you are talking about cycling through the phase transition of the alloy.

Answer: I have changed the description as you can see in line 150.

  • On line 219 you have two repeated explanations regarding the differences between your latent heat and the one reported in literature.

Answer: Thank you for reminding, I have deleted the repeated explanation in manuscript.

  • Figure 3: Maybe it is due to the pdf generation process, but if possible try to improve the quality of this figure.

Answer: I have attached a new version of fig.3 with high resolution, please check it.

  • On line 237: What can be the reason of the decrease of latent heat after cycling? TGA (thermogravimetric analysis) of the alloy up to 700ºC is required and could complete your work. Loss of Zn could be a reason.

Answer: I agree with you that the loss of Zn might be a reason for that. So I have added this explanation in the manuscript.

  • On line 247: “The coefficient of thermal expansion is an important parameter related to whether Mg–25%Cu–15%Zn 248 eutectic alloy can be used as phase change material, so the application of thermal expansion analysis and thermal expansion test is very important.” Why is this coefficient very important to determine if the material can be used as PCM?

Answer: If the volume or length of PCM changes with the temperature variation in the process of use, it will lead to leakage, which will influence the effect of PCM. Therefore, coefficient of thermal expansion is of importance in the application of PCM.

  • On line 253: “Then, the phase transition occurs at 442.0 °C, which is lower than 452.6 °C in the DSC measurements. This difference in onset temperature may be attributed to the different heating rate and calorimeter accuracies.” The different onset Temperature can be also related with the different sample masses used for each analysis (effect of sample mass in thermal properties is reported in literature).

Answer: Thanks for your reminding, I have added this explanation in my manuscript according to your suggestion in line 222.

  • The volume change from room temperature to 442ºC has been analyzed. What about the volume change due to the phase change (very important parameter overall taking into account that your PCM is sealed in a container), and density of liquid phase? It will be very helpful to have the volume expansion of the phase change. (recommendation)

Answer: Thanks for your suggestion, at first, I just measured the volume change from room temperature to the phase transition point. Your suggestion is very helpful for my follow-up research, and I will consider the change of liquid volume and density in the next research.

  • On line 289: “This value is higher than the result of Mg–51%Zn alloy reported 289 in the literature [24] and lower than the specific heat value of pure Mg [30].” Specify the values in the text.

Answer: Thanks for your advice, I have added the detailed information of the values in references. In addition, the value of pure Mg can be obtained by Cp(Mg)=0.89+4.58∗10-4T in ref. [30].

  • If the material is going to be applied as PCM the working temperature range will cover not only the solid phase, but also the liquid phase. Could you test the thermal conductivity of liquid phase? And do you have any information or have you tested the volume expansion due to the phase change? (recommendation)

Answer: Thanks for your precious advice, I will make a further study on the thermal conductivity of liquid phase in the nest research.

Round 2

Reviewer 1 Report

-The title has to be changed and English language has to be improved.

-Any phase diagram evolves with time and is updated hence, add the used one inside the paper to facilitate the reading please.

-What about the used protocols for the set-up of the sample inside the MEB chamber and for the related EDS (duration of the experiments...)?

-Additional experiments have to be performed again to allow a reliable interpretation of the obtained results and some complementary ones have to be done to consolidate or not these results. For instance, how many samples have been studied by using DSC device  (what about the reproducibility of the obtained results) ? Figure 3 shows the results obtained for one sample only that underwent the thermal cycling. No more information is given in the text. During the first cycle, the slight inflection observed both during the melting and the solidification steps shows that the chosen value for the heating and the cooling scanning rate is too high, so that at least the second peak is convoluted to the 1st one. Idem for Figure 2.  Figure 1 shows that there is a big difference between the first X-ray diffraction pattern at cycle n°0 and the one obtained after 100 cycles. Then, a slight visible change occurs between the pattern obtained for 400 cycles and the one obtained for 500 cycles. Thus, the material is not the same. It clearly appears that the material evolves with the thermal cycling. The MEB analyses would be more relevant with intermediate observations at these different thermal cycle number to compare them with the XRD analyses Anyway, the pictures confirm that the material is not the same anymore.

- The authors cannot claim that they obtained the right eutectic component without a comparison with the XRD pattern and the cristallographic structure established by the crystallographers. If it is the case, please show the superimposed patterns to confirm this claim.

-What are the heating and cooling scanning rates used the thermal cycling in the furnace ? 

-Why are the heating and cooling scanning rate values used for DSC experiments and for the thermal expansion ones ? Interpretation of the results cannot be definitely done. 

-l199: How to be sure that this is not an artefact due to the picture's contrast (which is not the same for both pictures)?

-l215-222: Not true. The value given in [29] and the only one is of 452 °C. The value obtained by the authors is 0.6°C higher maybe due to the errors related to the device but even the authors of [29] said they performed only 2 DSC experiments with two different values "too low" according to them...This is not reliable and sufficient information.. (and the paper was published in 1985...)

-l221-222: The authors cannot compare these values for the material studied in [24] is a binary one!!!

-l239-248: These arguments do not make sense here for they bring no useful information for the understanding of this work...otherwise, it is THE arguments for all works done in this field without explaining nothing.

Author Response

-The title has to be changed and English language has to be improved.

Answer: Thanks for your comment, I have realized the grammar error in the title and I corrected it.

-Any phase diagram evolves with time and is updated hence, add the used one inside the paper to facilitate the reading please.

Answer: Thanks for your reminding, I will add the phase diagram that I used in this manuscript, as you can see in line 240, page 7.

-What about the used protocols for the set-up of the sample inside the MEB chamber and for the related EDS (duration of the experiments...)?

Answer: Thanks for your reminding, I have added the expression about the process of conducting EPMA experiment as you can see in line 144. “In this study, EPMA were conducted in the spot mode with an accelerating voltage of 10 kV and a probe current of 10 nA. The beam diameter was in the range of 1 to 2 µm.”

-Additional experiments have to be performed again to allow a reliable interpretation of the obtained results and some complementary ones have to be done to consolidate or not these results. For instance, how many samples have been studied by using DSC device  (what about the reproducibility of the obtained results) ? Figure 3 shows the results obtained for one sample only that underwent the thermal cycling. No more information is given in the text. During the first cycle, the slight inflection observed both during the melting and the solidification steps shows that the chosen value for the heating and the cooling scanning rate is too high, so that at least the second peak is convoluted to the 1st one. Idem for Figure 2.  Figure 1 shows that there is a big difference between the first X-ray diffraction pattern at cycle n°0 and the one obtained after 100 cycles. Then, a slight visible change occurs between the pattern obtained for 400 cycles and the one obtained for 500 cycles. Thus, the material is not the same. It clearly appears that the material evolves with the thermal cycling. The MEB analyses would be more relevant with intermediate observations at these different thermal cycle number to compare them with the XRD analyses Anyway, the pictures confirm that the material is not the same anymore.

Answer: In fact, the cyclic test was repeated three times in each group, and the DSC test of parallel samples was carried out, and the error was within 3%, which proved that the reproducibility of the results was reliable. In the first cycle, the slight bending observed in the curves during melting and solidification is only a matter of baseline treatment, whether and when endothermic peaks appear in the range of RT-500 ℃ were our concerns. Fig. 1 and Fig. 2 show that the composition and phase transition enthalpy of the material system did change after many cycles, but compared with the engineering application potential of the material, the change of 500 cycles is acceptable.

- The authors cannot claim that they obtained the right eutectic component without a comparison with the XRD pattern and the cristallographic structure established by the crystallographers. If it is the case, please show the superimposed patterns to confirm this claim.

Answer: The crystal analysis of XRD was based on the comparison standard card and EDS. As for whether we have prepared the right eutectic alloy, we also combined the ternary alloy phase diagram, XRD, SEM-EDS and DSC results, finally, the eutectic alloy with melting point of 452.6 ℃ and transformation enthalpy of 177.5j/g was obtained.

-What are the heating and cooling scanning rates used the thermal cycling in the furnace ? 

Answer: Thanks for your reminding, both the heating and cooling scanning rates were 10 K/min, and I have added this information in the manuscript as you can see in line 129.

-Why are the heating and cooling scanning rate values used for DSC experiments and for the thermal expansion ones ? Interpretation of the results cannot be definitely done.

Answer: As mentioned in last question, the heating and cooling scanning rate used in the thermal cycling in the furnace was 10 K/min, while that used in DSC experiments and thermal expansion were 10K/min and 5K/min as you can see in line 150-154.  

-l199: How to be sure that this is not an artefact due to the picture's contrast (which is not the same for both pictures)?

Answer: EPMA pictures were obtained directly from EPMA without any treatment, picture’s contract is depend on the condition of testing for observing more clearly, and which has no influence on the authenticity of pictures. The images of α-Mg dendrites can be referred to the Fig. 2 in reference “Growth orientations and morphologies of α-Mg dendrites in Mg–Zn alloys. M.Y.Wang, et al. Scripta Materialia. 2012, 67, Pages: 629-632.”

-l215-222: Not true. The value given in [29] and the only one is of 452 °C. The value obtained by the authors is 0.6°C higher maybe due to the errors related to the device but even the authors of [29] said they performed only 2 DSC experiments with two different values "too low" according to them...This is not reliable and sufficient information.. (and the paper was published in 1985...)

Answer: Thank you for expressing your concerns, but all the values were obtained through our experiments without any embellishment or treatment. In addition, as a mature characterization technology, DSC has been widely used since 1980s. Therefore, I prefer to believe the authenticity of the data and results of this literature [29].

-l221-222: The authors cannot compare these values for the material studied in [24] is a binary one!!!

Answer: The introduction of reference [24] is just to indicate that on the basis of Mg / Zn binary alloy system, the enthalpy value and thermal conductivity of ternary eutectic alloy obtained by adding Cu were greatly improved.

-l239-248: These arguments do not make sense here for they bring no useful information for the understanding of this work...otherwise, it is THE arguments for all works done in this field without explaining nothing.

Answer: According to the comment from reviewer, I decided to delete this paragraph, as you can see in line 239-248.

Reviewer 4 Report

After the modifications made by the authors, I consider that the article has been improved and deserves to be published.

I would like to point out that my previous remark about the weight loss of the sample was simply a suggestion. The authors should not add this conclusion  (line 199) without at least referencing some previous work of other authors showing that Zn may be the cause of this loss. 

The decrease of latent heat after cycling might be attributed to the loss of Zn, be-199 cause Zn has a lower melting point

Author Response

Thanks for your comments, I have added the reference that can support the conjecture as you can see in line 199.